# ADVANCING MULTI-AGENT TRAFFIC SIMULATION VIA R1-STYLE REINFORCEMENT FINE-TUNING

**Muleilan Pei[1], Shaoshuai Shi[2]\*, Shaojie Shen[1]**
[1]Hong Kong University of Science and Technology  [2]Voyager Research, Didi Chuxing
`{mpei,eeshaojie}@ust.hk, shaoshuaics@gmail.com`

## ABSTRACT

Scalable and realistic simulation of multi-agent traffic behavior is critical for advancing autonomous driving technologies. Although existing data-driven simulators have made significant strides in this domain, they predominantly rely on supervised learning to align simulated distributions with real-world driving scenarios. A persistent challenge, however, lies in the distributional shift that arises between training and testing, which often undermines model generalization in unseen environments. To address this limitation, we propose SMART-R1, a novel R1-style reinforcement fine-tuning paradigm tailored for next-token prediction models to better align agent behavior with human preferences and evaluation metrics. Our approach introduces a metric-oriented policy optimization algorithm to improve distribution alignment and an iterative "SFT-RFT-SFT" post-training strategy that alternates between Supervised Fine-Tuning (SFT) and Reinforcement Fine-Tuning (RFT) to maximize performance gains. Extensive experiments on the large-scale Waymo Open Motion Dataset (WOMD) validate the effectiveness of this simple yet powerful R1-style training framework in enhancing foundation models. The results on the Waymo Open Sim Agents Challenge (WOSAC) showcase that SMART-R1 achieves state-of-the-art performance with an overall realism meta score of 0.7858, ranking first on the leaderboard at the time of submission.

## 1 INTRODUCTION

Simulating multi-agent traffic behaviors plays a pivotal role in ensuring the safety and reliability of autonomous driving systems. However, modeling realistic and scalable traffic behaviors remains highly challenging due to the inherent uncertainty and multi-modality of human driving. Traditional simulators that simply replay logged data lack reactive capability, while rule-based methods, such as the Intelligent Driver Model (IDM) (Treiber et al., 2000), depend on handcrafted heuristics and fail to capture the diversity and realism of human behavior. To overcome these limitations, recent research (Montali et al., 2023) has reframed realistic traffic simulation as a distribution matching problem, aiming to align the behavior distributions of simulated agents with those observed in real-world data. The guiding principle is that, as simulation quality improves, the distributions of learned simulated agent behaviors should assign higher likelihoods to real-world logged samples.

Early learning-based simulators largely adopted architectures designed for motion forecasting, such as encoder–decoder Transformers (Shi et al., 2024; Zhou et al., 2023). Yet, simulation fundamentally differs from trajectory prediction: it requires (i) a closed-loop setup, (ii) consistent scene-level multi-agent futures, and (iii) recovery of the underlying behavior distribution rather than simple trajectory imitation. These distinctions often cause motion predictors to perform poorly when adapted to simulation. More recent advances in high-performing simulators have primarily followed two directions: diffusion-based models (Jiang et al., 2024; Huang et al., 2024) and autoregressive models (Seff et al., 2023; Philion et al., 2024). Diffusion models can generate diverse multi-modal futures by modeling joint trajectory distributions, but they suffer from computational inefficiency and struggle to capture multi-agent interactions. In contrast, autoregressive approaches treat agent behaviors as discrete tokens under the Next-Token Prediction (NTP) paradigm (Wu et al., 2024; Zhou et al., 2024), generating interactive and realistic multi-agent behaviors through minimizing cross-entropy

---

*Corresponding Author.

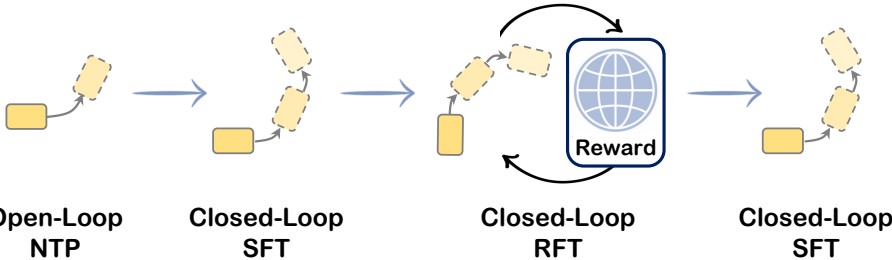

Figure 1: **Training pipeline of SMART-R1.** The solid square represents the agent state at the current timestep. The Open-Loop NTP predicts the next single-step state, while Closed-Loop SFT rolls out entire trajectories autoregressively to identify those closest to the ground truth. Both stages are optimized with per-token cross-entropy loss. In contrast, Closed-Loop RFT performs full rollouts but aligns trajectories with evaluation preferences through reward feedback and policy optimization.

loss against the distribution of logged tokens. However, because NTP models generate behaviors in an autoregressive manner, they are prone to encountering the covariate shift issue: small prediction errors accumulate during closed-loop simulation rollouts. Inspired by post-training strategies for Large Language Models (LLMs) (Ouyang et al., 2022), a Supervised Fine-Tuning (SFT) technique with Closest Among Top-K (CAT-K) rollouts (Zhang et al., 2025) has been proposed to mitigate compounding errors, significantly improving simulation realism.

Despite the remarkable performance of NTP models in traffic simulation, a fundamental limitation persists: the training objectives of current imitative or Behavior Cloning (BC) models are not explicitly aligned with ultimate behavior-characterizing goals of simulators, such as reducing collisions or off-road rates. These outcome metrics are often scalar, sparse, and non-differentiable, making them unsuitable as direct trainable loss functions for gradient-based optimization. Consequently, relying solely on BC or SFT is insufficient for realistic simulation agents.

To bridge this gap, we draw inspiration from cutting-edge Large Reasoning Models (LRMs) such as OpenAI-o1 (Jaech et al., 2024) and DeepSeek-R1 (Guo et al., 2025), which leverage Reinforcement Learning from Human Feedback (RLHF) or Reinforcement Fine-Tuning (RFT) to align model behaviors with specific preferences. In this regard, we introduce an R1-style RFT training paradigm for traffic simulation, as demonstrated in Figure 1. Specifically, we build on SMART (Wu et al., 2024), a strong open-loop NTP foundation model, and employ CAT-K rollouts (Zhang et al., 2025) for closed-loop SFT. In the subsequent RFT stage, we refrain from directly applying the Group Relative Policy Optimization (GRPO) (Shao et al., 2024) algorithm for preference alignment. Although GRPO eliminates the necessity for explicit value function approximations by utilizing normalized relative rewards within each group, its dependence on average rewards across multiple sampled rollouts introduces inherent sampling bias and undermines the reliability of advantage estimation for policy optimization (Zheng et al., 2025). Given the relatively predictable reward expectations in our task, we instead propose a Metric-oriented Policy Optimization (MPO) strategy, which exploits this prior knowledge to guide policy updates in RFT more efficiently and effectively.

Moreover, the conventional practice in LRMs typically involves leveraging SFT as a warm-up stage before RFT, which poses the issue of catastrophic forgetting (Chen et al., 2025), resulting in performance degradation. To address this concern, following the training pipeline of DeepSeek-R1, which alternates multiple rounds of SFT and RFT to refine reasoning performance, we introduce an additional closed-loop SFT phase after RFT. This yields an iterative "SFT-RFT-SFT" (R1-style) post-training scheme that balances optimization for metric-specific objectives with preservation of the behavioral distribution learned during SFT, thereby further enhancing overall simulation realism.

In summary, the primary contributions of this paper are as follows:

(1) We introduce SMART-R1, to our knowledge the first R1-style post-training paradigm for multi-agent traffic simulation, which combines SFT and RFT to better align simulated behaviors with human preferences and evaluation metrics.

(2) We develop a simple yet effective Metric-oriented Policy Optimization (MPO) strategy for RFT to reinforce the model towards targeted evaluation metrics.

(3) Our proposed post-fine-tuning pipeline, structured as "SFT-RFT-SFT", empowers SMART-R1 to achieve state-of-the-art performance on the 2025 Waymo Open Sim Agents Challenge, ranking first on the leaderboard at the time of submission.

## 2 RELATED WORK

**Traffic Behavior Modeling.** Accurately modeling the behaviors of traffic agents is a core component of autonomous driving systems. Early approaches represent behavior in a discrete and sparse decision space (e.g., lane-keeping or left-turning), but such abstractions are overly coarse to capture the diversity of real-world driving (Cunningham et al., 2015). More recent work instead models behavior as sequences of future position points (Pei et al., 2025b;a), and employs encoder–decoder architectures to generate continuous trajectories by framing the task as regression (Song et al., 2022; Shi et al., 2025). In contrast, emerging approaches characterize traffic behavior as a language modeling problem, where trajectories are discretized into motion tokens to form an action vocabulary. The resulting trajectories are then produced via next-token prediction in an autoregressive manner (Philion et al., 2024; Seff et al., 2023). This paradigm naturally captures multi-modal and interactive behaviors among agents, making it well-suited for realistic traffic simulation.

**Multi-agent Simulation.** Multi-agent traffic simulation is crucial for evaluating the robustness and safety of autonomous vehicles (Shi et al., 2026). Traditional simulators often depend on rule-based policies (Treiber et al., 2000) for traffic participants, which fail to reproduce the complexity and realism of human driving. To address this, data-driven approaches have been proposed to enable scalable and realistic simulation. Early methods repurpose trajectory prediction models for simulation (Shi et al., 2024; Zhou et al., 2023); however, they lack explicit interaction modeling and exhibit poor stability in closed-loop rollouts. Recent advances in generative modeling have mainly followed two directions: diffusion-based methods (Jiang et al., 2024; Hu et al., 2024) and autoregressive methods (Wu et al., 2024; Zhou et al., 2024; Peng et al., 2024). While diffusion models can capture multi-modality, they suffer from computational inefficiency and degrade in long-horizon scenarios. Consequently, research momentum has shifted toward autoregressive models. In line with these trends, we adopt the next-token prediction paradigm for the multi-agent traffic simulation task.

**Reinforcement Fine-Tuning.** Post-training is widely utilized in LLMs to enhance the model alignment with human preferences. Early efforts primarily concentrate on SFT, while more recent work explores RFT to improve reasoning capabilities. For example, OpenAI-o1 leverages Proximal Policy Optimization (PPO) (Schulman et al., 2017) for RLHF. Subsequently, Direct Preference Optimization (DPO) (Rafailov et al., 2023) is proposed to simplify the RLHF process by eliminating the need for training a reward model. More recently, DeepSeek-R1 introduces GRPO (Shao et al., 2024) within a multi-stage training pipeline that alternates between SFT and RFT to progressively refine reasoning performance. Building on this trend, a series of R1-style approaches have been proposed to extend RFT-based alignment to domains such as vision reasoning (Huang et al., 2025; Tan et al., 2025) and trajectory planning (Li et al., 2025; Tang et al., 2025). However, most prior works employ RFT-empowered LLMs as auxiliary components for downstream tasks, rather than integrating RFT directly into the downstream domain itself. In contrast, to the best of our knowledge, our work is the first to apply R1-style RFT directly to the domain of multi-agent traffic simulation.

## 3 METHODOLOGY

### 3.1 PRELIMINARIES

**Problem Formulation.** For multi-agent traffic simulation, the problem is often framed as learning a policy $\pi_\theta(S_t|S_{<t}, \mathcal{C})$, where $S_t = \{s_t^i\}_{i=1}^N$ denotes the state of $N$ agents in the scene at time step $t$, $S_{<t} = \{s_{<t}^i\}_{i=1}^N$ is the historical states, and $\mathcal{C}$ represents the contextual information such as features from High-Definition (HD) maps. The trainable model parameters are denoted by $\theta$. The objective is to learn a joint distribution of agent behaviors within the set of input information over a time horizon $\tau$:

$$P(S_{1:\tau}|S_0, \mathcal{C}) = \prod_{t=1}^{\tau} \pi_\theta(S_t|S_{<t}, \mathcal{C}). \tag{1}$$

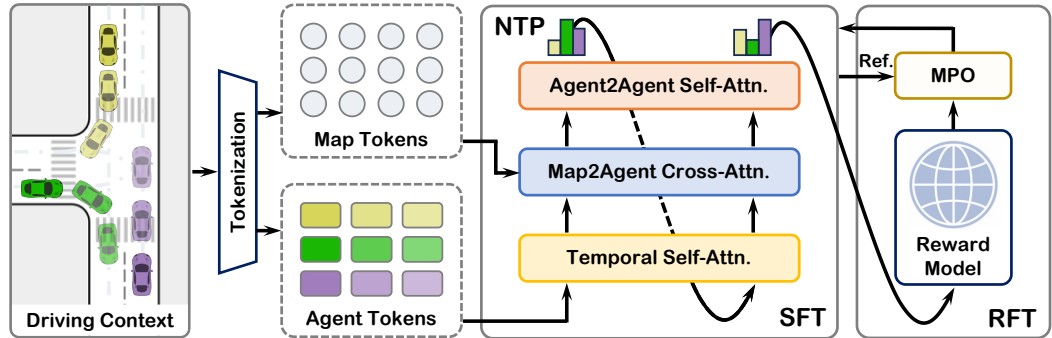

Figure 2: **Framework of SMART-R1.** The model first tokenizes the driving scene context into agent motion tokens and map tokens. In the BC pretraining stage, the model is optimized under the standard NTP paradigm. During the SFT stage, CAT-K rollouts are used to refine the model in a closed-loop setting. Finally, in the RFT stage, the proposed MPO algorithm aligns the policy with target evaluation metrics, further improving simulation realism.

**NTP Models for Traffic Simulation.** To model agent behaviors within the Next-Token Prediction (NTP) framework, we often preprocess the driving context by discretizing and clustering continuous agent trajectories into a motion token vocabulary $\mathcal{V} = \{m_k\}_{k=1}^{|\mathcal{V}|}$, comprising $|\mathcal{V}|$ template motion tokens $m_k$. Accordingly, the policy $\pi_\theta(S_t|S_{<t}, \mathcal{C})$ can be reformulated as predicting a categorical distribution over the indices of the motion token vocabulary at each time step $t$, i.e.,

$$\pi_\theta(S_t|S_{<t}, \mathcal{C}) = \prod_{i=1}^{N} \pi_\theta(k_t^i|S_{<t}^*, \mathcal{C}^*), \tag{2}$$

where $X^*$ denotes the tokenized format of the inputs $X$.

**Closed-Loop SFT.** To further mitigate the covariate shift introduced by open-loop behavior cloning, the CAT-K rollout strategy (Zhang et al., 2025) can be leveraged for closed-loop SFT in the traffic simulation task. Specifically, after obtaining the pre-trained foundation model, we autoregressively unroll the policy to generate tokenized trajectories and construct recovery motion indices to stay close to the ground truth. This closed-loop SFT approach effectively reduces covariate shift in NTP models and leads to improved overall performance.

## 3.2 FRAMEWORK OVERVIEW

The overall architecture of our proposed SMART-R1 is illustrated in Figure 2. It follows a multi-stage training paradigm consisting of Behavior Cloning (BC) pretraining, closed-loop SFT, and RFT. Given a real-world scenario, following prior work (Wu et al., 2024; Zhang et al., 2025), we first discretize continuous trajectories and map polylines into agent motion tokens and map tokens using the K-disks clustering algorithm (Philion et al., 2024). These tokens are then processed by a stack of attention layers: a temporal self-attention layer to capture sequential dependencies in agent motions, a map-to-agent cross-attention layer to incorporate map-relevant information, and an agent-to-agent self-attention layer to model interactions among traffic participants for generating joint multi-agent behaviors. The outputs from these modules produce the next-token logits. In the BC pretraining stage, the model is optimized via token-level classification. The resulting pretrained base model is subsequently refined through closed-loop SFT, where CAT-K rollouts are employed to mitigate the covariate shift inherent in BC. Finally, in order to align the model with metric preferences, we introduce an RFT stage and develop a simple yet effective Metric-oriented Policy Optimization (MPO) algorithm for refinement, further enhancing overall simulation realism.

## 3.3 METRIC-ORIENTED POLICY OPTIMIZATION

Considering that both the open-loop pre-trained model and the closed-loop SFT stage are trained using cross-entropy loss to learn the distribution of logged data, they lack direct optimization for

---

**Algorithm 1** Metric-oriented Policy Optimization (MPO)

---

**Input:** Initial policy $\pi_{\theta_{\text{init}}}$.
**Output:** Optimized RFT policy $\pi_{\theta_{\text{rft}}}$.
 1: Initialize the reference model $\pi_{\theta_{\text{ref}}} \leftarrow \pi_{\theta_{\text{init}}}$.
 2: Initialize the policy model $\pi_\theta \leftarrow \pi_{\theta_{\text{init}}}$.
 3: **for** iteration $= 1, 2, \ldots$ **do**
 4:  Roll out multi-agent trajectories in the scene via $\pi_\theta$.
 5:  Compute the reward function $r$ for the simulation outcomes using the Realism Meta metric.
 6:  Compute the simplified generalized advantage estimation $\mathcal{A}$ via Equation 3.
 7:  Compute the unbiased estimation of per-token KL divergence $\mathcal{D}_{\text{KL}}[\pi_\theta || \pi_{\theta_{\text{ref}}}]$ via Equation 4.
 8:  Update policy $\pi_\theta$ by minimizing $\mathcal{L}_{\text{MPO}}$ via Equation 5.
 9: **end for**

---

task-specific evaluation metrics. To bridge this gap, we introduce RFT to explicitly align the pre-trained NTP model with a designated target metric.

We begin by formulating the behavior simulation problem as a Markov Decision Process (MDP), where each tokenized motion corresponds to an element in the action space. The policy $\pi_\theta$ deterministically transitions each agent to its next state by selecting the associated motion token. The reward model $r$ is defined by the Realism Meta metric computed through the official evaluation protocol.

Building upon the MDP formulation, we propose the Metric-oriented Policy Optimization (MPO) to maximize the cumulative Realism Meta score. As illustrated in Figure 3, for each traffic scenario, we autoregressively roll out complete trajectories for all agents and evaluate the resulting simulation using the official Realism Meta metric. This metric serves as a reward model, assigning a scalar score $r$ to assess the quality of the generated behaviors. Note that we can also apply other preferences to reinforce the alignment. Unlike GRPO (Shao et al., 2024), which relies on generating a group of completions for comparison, our approach simplifies the estimation of the advantage function, as in our specific task, we have the expected reward value for the model to achieve. Specifically, we compute the Generalized Advantage Estimation (GAE), denoted $\mathcal{A}$, as follows:

$$\mathcal{A} = r - \alpha, \tag{3}$$

where $\alpha$ is an empirical threshold used to normalize the reward signal and facilitate relative evaluation for efficient policy optimization.

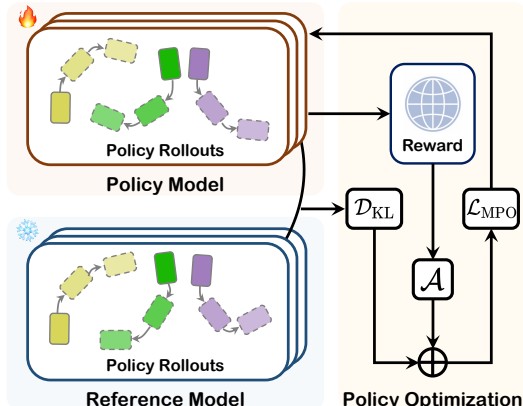

Figure 3: Pipeline of the Metric-oriented Policy Optimization (MPO).

We also incorporate a per-token KL divergence penalty between the updated policy $\pi_\theta$ and a reference model $\pi_{\theta_{\text{ref}}}$ using the following unbiased estimator:

$$\mathcal{D}_{\text{KL}}[\pi_\theta || \pi_{\theta_{\text{ref}}}] = \frac{\pi_{\theta_{\text{ref}}}}{\pi_\theta} - \log \frac{\pi_\theta}{\pi_{\theta_{\text{ref}}}} - 1. \tag{4}$$

The final training objective for MPO is then given by:

$$\mathcal{L}_{\text{MPO}} = -(\frac{\pi_\theta}{\overline{\pi}_\theta} \mathcal{A} - \beta \mathcal{D}_{\text{KL}}[\pi_\theta || \pi_{\theta_{\text{ref}}}]), \tag{5}$$

where $\overline{\pi}_\theta$ is a no-gradient copy of the policy, and $\beta$ is a tunable coefficient balancing the advantage term against the KL penalty. The complete pipeline of MPO is summarized in Algorithm. 1.

### 3.4 R1-Style Training Paradigm

Since models fine-tuned with RFT risk deviating from the original distribution learned during pre-training and SFT, leading to catastrophic forgetting (Chen et al., 2025), we draw inspiration from

the multi-stage post-training pipeline of DeepSeek-R1 (Guo et al., 2025) and introduce an iterative "SFT–RFT–SFT" strategy. Starting from a pretrained base model, we first apply CAT-K rollouts for closed-loop SFT to mitigate covariate shift and stabilize the policy. We then employ the proposed MPO technique to further refine the model by explicitly aligning it with evaluation metrics. Finally, we conduct an additional round of closed-loop SFT to restore fidelity to the logged data distribution, thereby enhancing simulation performance.

Empirically, this R1-style paradigm consistently outperforms standalone SFT or RFT (see Table 3). We attribute its effectiveness to striking a balance between optimizing for metric-driven objectives and preserving generalization from real-world data, and thus mitigating forgetting and improving the realism of simulated traffic behaviors.

### 3.5 TRAINING OBJECTIVES

Given our proposed framework, which consists of three phases, the training process is decomposed into three stages with distinct training objectives. In the open-loop pretraining stage, we optimize the NTP task by minimizing the cross-entropy loss between the ground-truth token distribution and the prediction. Given the corresponding ground-truth index, denoted as $\tilde{k}$, the training objective for policy rollouts can be formally defined as follows:

$$\mathcal{L}_{\text{base}} = -\sum_{t=1}^{\tau}\sum_{i=1}^{N} \log\left(\pi_\theta(\tilde{k}_t^i|S_{<t}^*, \mathcal{C}^*)\right). \tag{6}$$

In the closed-loop SFT stage, we perform CAT-K rollouts (Zhang et al., 2025) to obtain a sequence of motion tokens and identify the motion index $\hat{k}$ that is closest to the ground-truth trajectories. The model is again optimized using cross-entropy loss, with Equation 6 applied by replacing $\tilde{k}$ with $\hat{k}$. Finally, in the RFT stage, we employ the $\mathcal{L}_{\text{MPO}}$, as defined in Section 3.3, to directly align the policy with target evaluation metrics.

## 4 EXPERIMENTS AND RESULTS

### 4.1 EXPERIMENTAL SETUP

**Real-World Dataset.** We train our model on the large-scale Waymo Open Motion Dataset (WOMD) (Ettinger et al., 2021) and evaluate it on the Waymo Open Sim Agents Challenge (WOSAC) benchmark (Montali et al., 2023). WOMD contains approximately 487k training scenarios, 44k validation scenarios, and 45k test scenarios. Each scenario spans 9.1 seconds at 10 Hz. Given 10-step historical tracks, the current state, and the corresponding HD map, the task is targeted to simulate 32 realistic joint 8-second futures (including 80-step centroid coordinates and heading of the objects' boxes) for all the agents (up to 128) in the scene.

**Evaluation Metrics.** We adopt the official WOSAC evaluation metrics, which frame simulation as a distribution matching problem: assessing how well the density induced by simulated rollouts assigns likelihood to real-world trajectories. The primary evaluation criterion is the Realism Meta metric, a weighted combination of three sub-metrics, including Kinematic, Interactive, and Map Adherence, each comprising several finer-grained measures (see Table 2). In addition, we report the minimum Average Displacement Error (minADE), a standard open-loop prediction metric that serves as a complementary reference for motion forecasting accuracy. More details can be found in the official WOSAC documentation (Montali et al., 2023) and Appendix A.1.

**Implementation Details.** We choose SMART (Wu et al., 2024) as our baseline NTP model. For both the open-loop NTP and closed-loop SFT training processes, we follow the official implementations of Zhang et al. (2025) to ensure fair comparison. Specifically, we pretrain the foundation autoregressive model SMART-tiny (7M parameters) for 64 epochs to obtain the base model. We then perform post-training, including SFT and RFT. Instead of conducting a 32-epoch SFT as the original implementation of Zhang et al. (2025), we split the post-training into three phases: (1) an initial SFT stage of 16 epochs, (2) RFT with our proposed MPO algorithm, and (3) a final SFT stage of 16 epochs. For MPO, we set the empirical reward threshold to $\alpha = 0.77$ and the KL penalty coefficient to $\beta = 0.04$. Note that no model ensembling or post-processing tricks are employed, as

Table 1: **Performance comparison on the 2025 Waymo Open Sim Agents Challenge (WOSAC) leaderboard.** The Realism Meta metric serves as the official ranking metric, shaded in gray. The best and second-best results are highlighted in **bold** and underline, respectively. "*" indicates results reproduced using the official implementation and re-evaluated under the 2025 version of metrics.

| Method | Realism Meta metric ↑ | Kinematic metrics ↑ | Interactive metrics ↑ | Map-based metrics ↑ | min ADE ↓ | # model params |
|---|---|---|---|---|---|---|
| SMART* (Wu et al., 2024) | 0.7814 | 0.4854 | 0.8089 | 0.9153 | 1.3931 | 7M |
| SimFormer (Wang et al., 2025) | 0.7820 | 0.4920 | 0.8060 | 0.9167 | 1.3221 | 7M |
| UniMM (Lin et al., 2025) | 0.7829 | 0.4914 | 0.8089 | 0.9161 | 1.2949 | 4M |
| comBOT (Rossert et al., 2025) | 0.7837 | 0.4899 | 0.8102 | 0.9175 | 1.3687 | N.A. |
| RLFTSim (Ahmadi et al., 2025) | 0.7844 | 0.4893 | **0.8128** | 0.9164 | 1.3470 | 7M |
| CLSFT (Zhang et al., 2025) | 0.7846 | 0.4931 | 0.8106 | 0.9177 | 1.3065 | 7M |
| TrajTok (Zhang et al., 2026) | 0.7852 | 0.4887 | 0.8116 | **0.9207** | 1.3179 | 10M |
| **SMART-R1 (Ours)** | **0.7858** | **0.4944** | 0.8110 | 0.9201 | **1.2885** | 7M |

Table 2: Performance comparison among the SMART series across all sub-metrics on the test split.

| Model | Kinematic metrics | | | | Interactive metrics | | | Map-based metrics | | |
|---|---|---|---|---|---|---|---|---|---|---|
| | Lin. Speed ↑ | Lin. Acc. ↑ | Ang. Speed ↑ | Ang. Acc. ↑ | Dist.to Obj. ↑ | Collision ↑ | T.T.C. ↑ | Dist. to Road Edge ↑ | Offroad ↑ | Traf. Light Violation ↑ |
| SMART-base | 0.3719 | 0.3984 | 0.5156 | 0.6557 | 0.3877 | 0.9693 | 0.8290 | 0.6757 | 0.9501 | 0.9809 |
| SMART-SFT | **0.3868** | 0.4066 | 0.5203 | 0.6588 | **0.3922** | 0.9702 | 0.8302 | 0.6814 | 0.9524 | 0.9805 |
| SMART-R1 | 0.3867 | **0.4078** | **0.5215** | **0.6614** | 0.3917 | **0.9709** | **0.8303** | **0.6820** | **0.9554** | **0.9817** |

trajectories must be sampled autoregressively for the simulation task. In addition, no external data beyond WOMD is used during training.

## 4.2 QUANTITATIVE RESULTS

We compare our proposed SMART-R1 against other state-of-the-art methods on the challenging 2025 WOSAC benchmark. As shown in Table 1, we report results from several top-ranking published approaches evaluated under the 2025 version of the metrics. Our SMART-R1 achieves first place on the leaderboard, attaining a Realism Meta metric of 0.7858, underscoring its ability to simulate multi-agent future behaviors that closely match real-world scenarios. Moreover, SMART-R1 achieves leading performance in both kinematic metrics and minADE, highlighting its capacity to effectively capture agent motion profiles while delivering the best open-loop prediction accuracy.

To further analyze its effectiveness, we provide a detailed comparison within the SMART series, including the base model SMART-base, the closed-loop fine-tuned SMART-SFT, and our proposed SMART-R1, across all sub-metrics, as listed in Table 2. The results show that RFT consistently improves performance across most evaluation aspects, with particularly notable gains on safety-critical metrics such as collisions, off-road rate, and traffic light violations, whose objectives are difficult to optimize with supervised learning, as these outcomes are often non-differentiable scalars. These findings demonstrate that incorporating RFT not only enhances behavioral interaction and safety but also improves overall simulation realism, all without introducing additional model parameters.

## 4.3 ABLATION STUDIES

We conduct thorough ablation studies to validate the effectiveness of key components in SMART-R1. Due to the high computational cost of evaluating on the full validation set, we follow Zhang et al. (2025) that performs efficient ablations on a 2% validation split, which has been verified to yield consistent conclusions with the full set.

**Effect of the R1-Style Training Paradigm.** We first investigate the impact of the proposed post-training paradigm. As shown in Table 3, the post-fine-tuning strategy consistently improves the simulation quality over the BC-pretrained model. Specifically, inserting an RFT phase after the initial SFT yields clear gains in the Realism Meta metric, as RFT directly optimizes toward the evaluation

Table 3: Effect of post-fine-tuning strategies.

| 1$^{st}$ SFT | RFT | 2$^{nd}$ SFT | Realism Meta metric ↑ | min ADE ↓ |
|---|---|---|---|---|
| | | | 0.7725 | 1.4335 |
| ✓ | | | 0.7734 | **1.3982** |
| ✓ | | ✓ | 0.7730 | 1.4188 |
| ✓ | ✓ | | 0.7740 | 1.4216 |
| ✓ | ✓ | ✓ | **0.7746** | 1.4194 |

Table 4: Effect of different RFT policies.

| Method | Realism Meta metric ↑ | min ADE ↓ |
|---|---|---|
| SMART-SFT | 0.7734 | **1.3982** |
| w/ PPO | 0.7611 | 1.5126 |
| w/ DPO | 0.7682 | 1.5465 |
| w/ GRPO | 0.7701 | 1.4553 |
| w/ MPO | **0.7740** | 1.4216 |

Table 5: Effect of empirical threshold.

| Threshold $\alpha$ | Realism Meta metric ↑ | min ADE ↓ |
|---|---|---|
| 0.76 | 0.7735 | 1.4348 |
| 0.77 | **0.7740** | **1.4216** |
| 0.78 | 0.7727 | 1.4492 |

Table 6: Effect of KL regularization.

| KL coefficient $\beta$ | Realism Meta metric ↑ | min ADE ↓ |
|---|---|---|
| 0.004 | 0.7732 | 1.4387 |
| 0.04 | **0.7740** | 1.4216 |
| 0.4 | 0.7724 | **1.4215** |

objective. Notably, applying two consecutive SFT phases without RFT degrades performance, underscoring the importance of the intermediate RFT stage. In contrast, applying an additional SFT phase after RFT further boosts performance, confirming the effectiveness of our R1-style strategy. This finding aligns with Chen et al. (2025), which notes that models trained with SFT followed by RFT often suffer from catastrophic forgetting. In summary, SFT restores the logged data distribution, while RFT shifts the policy toward the metric objective; the "SFT-RFT-SFT" iterative training pipeline balances these two objectives, yielding higher overall realism.

**Effect of the Policy Optimization.** We further compare several representative policy optimization methods, including PPO, DPO, GRPO, and our proposed MPO. Using SMART with one-stage SFT as the baseline, we observe in Table 4 that PPO, DPO, and GRPO all degrade performance. PPO suffers from unstable training due to its actor–critic structure, where the value model is often hard to optimize. DPO and GRPO avoid value approximation but rely on sampled rollouts, which can bias policy updates toward suboptimal candidate behaviors. In contrast, our metric-oriented policy directly leverages the relatively predictable reward expectation as prior knowledge, effectively guiding optimization. A key factor influencing the performance of MPO is the choice of the empirical threshold $\alpha$. As shown in Table 5, both higher and lower thresholds degrade results, since higher values provide too few positive rewards, while lower values make the criterion too lenient. It illustrates that policies are encouraged when exceeding the threshold and penalized otherwise. Therefore, with the baseline reward averaging around 0.77, setting $\alpha = 0.77$ yields the best simulation performance.

**Effect of the KL Regularization.** We finally study the impact of the KL regularization during RFT. Table 6 shows that when the KL penalty coefficient $\beta$ is too small, the policy deviates excessively from the reference model, leading to degraded performance by discarding prior knowledge from BC pretraining and SFT. Conversely, when $\beta$ is too large, the reference model dominates, limiting improvements from the reward signal and overlooking safety-critical aspects such as collision and off-road rates. A balanced choice of $\beta$ preserves the reference distribution while allowing effective metric-driven optimization.

## 4.4 QUALITATIVE RESULTS

We qualitatively evaluate SMART-R1 on the WOSAC validation split, focusing on a challenging intersection scenario. As shown in Figure 4, we visualize future trajectories across multiple timesteps. Two distinct rollouts (R1 and R2) for the same scene generated by our model are presented, which, while differing from the logged data, remain realistic and highlight the diversity of SMART-R1.

In Rollout R1 (second row of Figure 4), the model exhibits conservative behavior. The upper ellipse shows a vehicle slowing down to yield to a pedestrian crossing the road before making a right turn, while the lower ellipse highlights both the white car and the blue ego car (performing an unprotected left turn) decelerating to negotiate passage through the intersection. By contrast, Rollout R2 (final

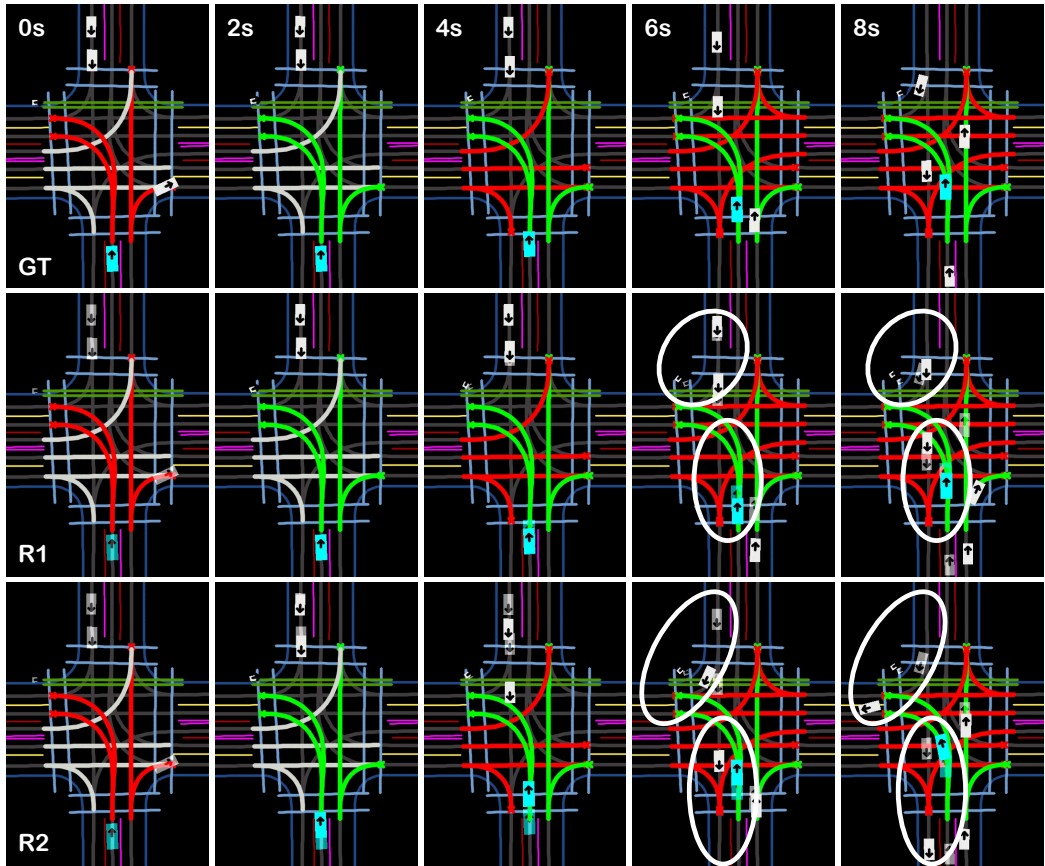

Figure 4: **Simulation results of SMART-R1.** The top row shows the ground-truth (GT) scenario, while the two rows below illustrate two representative rollouts generated by our model for the same scene. The blue box denotes the ego vehicle, and white boxes denote other traffic participants, including cars and pedestrians. Transparent boxes indicate ground-truth agent positions, whereas solid boxes represent simulated agents. Green lanes indicate traversable paths under a green light, while red lanes denote restricted paths.

row) demonstrates a more aggressive strategy. The upper ellipse shows the car executing a fast right turn, while the lower ellipse depicts the straight-moving car quickly passing through the intersection, with the ego car also advancing more rapidly than in the logged data.

These results demonstrate that SMART-R1 can not only imitate demonstrated behaviors but also generate diverse, plausible, and realistic traffic behaviors across different rollouts, effectively capturing both conservative and aggressive driving styles. Additional qualitative results are presented in Appendix A.3, with corresponding simulation videos available in the supplementary material.

## 5 CONCLUSION

In this paper, we introduce SMART-R1, to our knowledge, the first R1-style reinforcement fine-tuning paradigm for the next-token prediction model in the domain of traffic simulation. SMART-R1 incorporates a Metric-oriented Policy Optimization (MPO) algorithm to explicitly guide the model toward desired objectives, and employs an iterative "SFT–RFT–SFT" post-training strategy to effectively leverage SFT in conjunction with RFT. Experimental results demonstrate that our approach substantially enhances the overall simulation realism and achieves state-of-the-art performance on the 2025 Waymo Open Sim Agents Challenge with a Realism Meta metric of 0.7858 and a minADE of 1.2885, highlighting the promise of RFT for advancing multi-agent traffic simulation toward more realistic and safety-critical applications.

ACKNOWLEDGMENTS

This work was supported in part by the Hong Kong Ph.D. Fellowship Scheme and in part by the HKUST-DJI Joint Innovation Laboratory. The authors thank the Program Chairs for their support and assistance, as well as Lijun Yu, Shizhe Diao, Jun Cen, Xiaogang Jia, Jieqi Shi, Tin-Hing Wong, Xuesong Chen, Chen Shi, Muxin Liu, Ruixiang Li, Can Wang, and Jianfei Yang.

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

# A  APPENDIX

## A.1  2025 VERSION OF WOSAC METRICS

In this section, we provide a detailed overview of the evaluation metrics used in the 2025 Waymo Open Sim Agents Challenge (WOSAC) (Montali et al., 2023). The WOSAC metrics are updated annually, with adjustments made to the computation methods, the included sub-metrics, and their respective weights. Herein, we summarize the 2025 version of the metrics. Compared to the previous year, the most significant changes are: (i) the introduction of a new traffic light violation metric, and (ii) a new filtering mechanism for vehicles in the time-to-collision metric calculation. Below, we detail the definitions and the corresponding weights $w$ of the sub-metrics within the three main categories: kinematic, interactive, and map-based metrics.

(1) Kinematic metrics evaluate agent motion features and include four sub-metrics:

- Linear speed ($w = 0.05$): magnitude of the displacement in $(x, y, z)$ at each step.
- Linear acceleration ($w = 0.05$): signed change in speed magnitude between consecutive steps.
- Angular speed ($w = 0.05$): signed change in yaw between consecutive steps.
- Angular acceleration ($w = 0.05$): signed change in angular speed between consecutive steps.

(2) Interactive metrics assess how agents interact with each other and include three sub-metrics:

- Collision indication ($w = 0.25$): boolean indicating whether an object collides with another agent.
- Distance to nearest object ($w = 0.10$): minimum distance from the agent to any other agent. Negative values indicate overlap (collision).
- Time to collision ($w = 0.10$): predicted time (in seconds) before the agent collides with another agent, assuming constant velocity. This metric is filtered for vehicles only.

(3) Map-based metrics capture how agents behave relative to the environment and include three sub-metrics in the 2025 version:

- Off-road indication ($w = 0.25$): boolean indicating whether the agent is outside the drivable region of the map.
- Distance to road edge ($w = 0.05$): minimum distance to the boundary of the drivable region. Negative values indicate the agent is inside the road, positive values indicate otherwise.
- Traffic light violation ($w = 0.05$): boolean indicating whether a vehicle runs a red light. This metric is filtered for vehicles only.

Finally, the Realism Meta metric is computed as a weighted combination of the above sub-metrics, providing a comprehensive measure of overall simulation realism and performance.

## A.2  ADDITIONAL QUANTITATIVE RESULTS

In this section, we present detailed quantitative results from the top-ranking published approaches listed in Table 1, evaluated across all sub-metrics of the WOSAC leaderboard to enable a comprehensive comparison. All metrics follow a higher-is-better criterion. As shown in Table 7, our proposed SMART-R1 not only achieves the best performance in the overall Realism Meta metric but also ranks first or second across most individual sub-metrics.

Notably, TrajTok (Zhang et al., 2026) performs relatively well on map-based metrics. This can be attributed to its trajectory tokenizer, which integrates rule-based priors. However, such design choices make its generated trajectories less human-like, which explains its weaker performance on kinematic metrics. Meanwhile, comBot (Rossert et al., 2025), an ensemble-based method, faces scalability concerns, particularly for large-scale real-time simulation.

In summary, SMART-R1 delivers consistently competitive results across nearly all sub-metrics and achieves the top score in the overall Realism Meta evaluation, underscoring its effectiveness in producing realistic and high-fidelity traffic simulations.

Table 7: **Performance comparison across all sub-metrics on the 2025 WOSAC leaderboard**. The Realism Meta metric serves as the official ranking metric. The best and second-best results are highlighted in **bold** and underline, respectively. "*" indicates results reproduced using the official implementation and re-evaluated under the 2025 version of metrics.

| Model | Realism Meta metric ↑ | Kinematic metrics | | | | Interactive metrics | | | Map-based metrics | | |
|---|---|---|---|---|---|---|---|---|---|---|---|
| | | Lin. Speed ↑ | Lin. Acc. ↑ | Ang. Speed ↑ | Ang. Acc. ↑ | Dist.to Obj. ↑ | Collision ↑ | T.T.C. ↑ | Dist. to Road Edge ↑ | Offroad ↑ | Traf. Light Violation ↑ |
| SMART* | 0.7814 | 0.3719 | 0.3984 | 0.5156 | 0.6557 | 0.3877 | 0.9693 | 0.8290 | 0.6757 | 0.9501 | 0.9809 |
| SimFormer | 0.7820 | 0.3857 | 0.4066 | 0.5191 | 0.6565 | **0.3922** | 0.9619 | 0.8301 | 0.6814 | 0.9514 | 0.9785 |
| UniMM | 0.7829 | 0.3836 | 0.4160 | 0.5168 | 0.6491 | 0.3910 | 0.9680 | 0.8293 | 0.6791 | 0.9505 | 0.9811 |
| comBOT | 0.7837 | 0.3802 | 0.4033 | 0.5181 | 0.6580 | 0.3906 | 0.9703 | 0.8297 | 0.6790 | 0.9525 | 0.9630 |
| RLFTSim | 0.7844 | 0.3793 | 0.4026 | 0.5184 | 0.6568 | 0.3894 | **0.9755** | 0.8293 | 0.6779 | 0.9512 | 0.9809 |
| CLSFT | 0.7846 | **0.3868** | 0.4066 | 0.5203 | 0.6588 | **0.3922** | 0.9702 | 0.8302 | 0.6814 | 0.9524 | 0.9805 |
| TrajTok | 0.7852 | 0.3823 | 0.3988 | 0.5194 | 0.6542 | 0.3921 | 0.9742 | 0.8246 | **0.6845** | **0.9555** | **0.9829** |
| SMART-R1 | **0.7858** | 0.3867 | **0.4078** | **0.5215** | **0.6614** | 0.3917 | 0.9709 | **0.8303** | 0.6820 | 0.9554 | 0.9817 |

## A.3 ADDITIONAL QUALITATIVE RESULTS

We present additional qualitative results on the WOSAC validation split. Figure 5 illustrates a challenging U-turn scenario, where the simulated ego vehicle (blue box in the ellipse) successfully completes the U-turn in exact agreement with the logged trajectory. Figure 6 highlights an intersection with multi-agent negotiation, focusing on two white vehicles (ellipse) engaged in a strong interaction. We provide three rollouts demonstrating distinct outcomes: (R1) conservative waiting with mutual yielding at the intersection, (R2) the left-turning vehicle yielding to the through-going vehicle, and (R3) the through-going vehicle yielding to the left-turning vehicle. Figure 7 presents another intersection, featuring negotiation between the ego vehicle (blue box) and a traffic participant (white box). SMART-R1 generates three realistic behaviors: (R1) replication of the logged trajectory, (R2) a conservative maneuver, and (R3) an aggressive maneuver. The corresponding simulation videos are provided in the supplementary material.

These visualizations further highlight the ability of SMART-R1 to simulate diverse, plausible, and realistic multi-agent traffic behaviors, underscoring its strong capability in capturing complex interactions in real-world scenarios.

## A.4 DECLARATION OF LLM USAGE

Large Language Models (LLMs) were employed solely to assist with language refinement and manuscript polishing, including grammar correction and improvements in clarity and readability. All scientific concepts, contributions, methodological developments, experimental designs, data analyses, and conclusions are the original work of the authors. The authors have carefully reviewed and validated all content and take full responsibility for the accuracy and integrity of this manuscript.

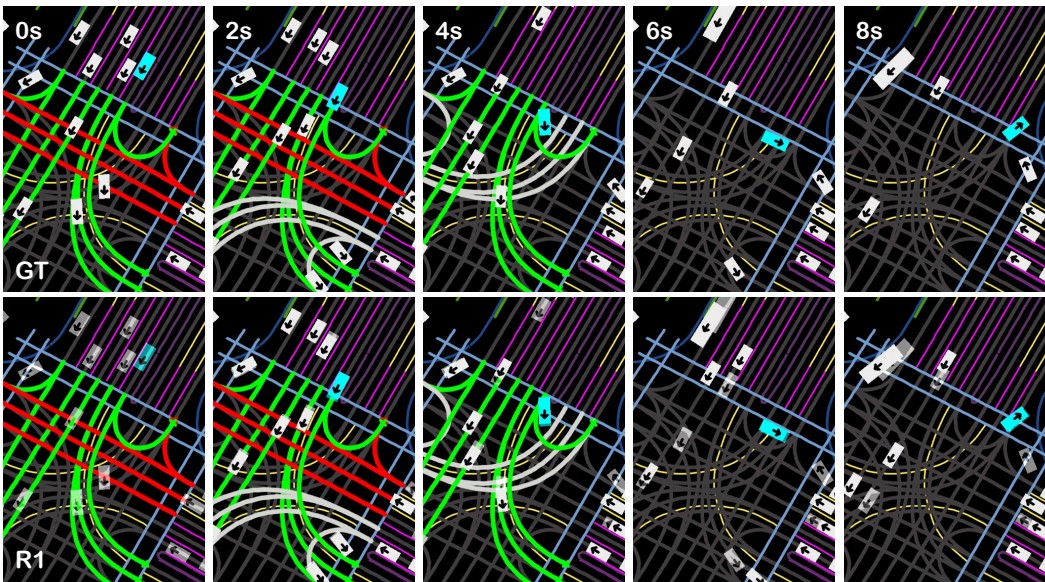

Figure 5: **Simulation results of SMART-R1 in a U-turn scenario.** The top row shows the ground-truth (GT) scenario. The blue box denotes the ego vehicle, and white boxes denote other traffic participants, including cars and pedestrians. Transparent boxes indicate ground-truth agent positions, whereas solid boxes represent simulated agents. Green lanes indicate traversable paths under a green light, while red lanes denote restricted paths. The visualization depicts the simulated ego vehicle completing the U-turn in exact agreement with the logged trajectory.

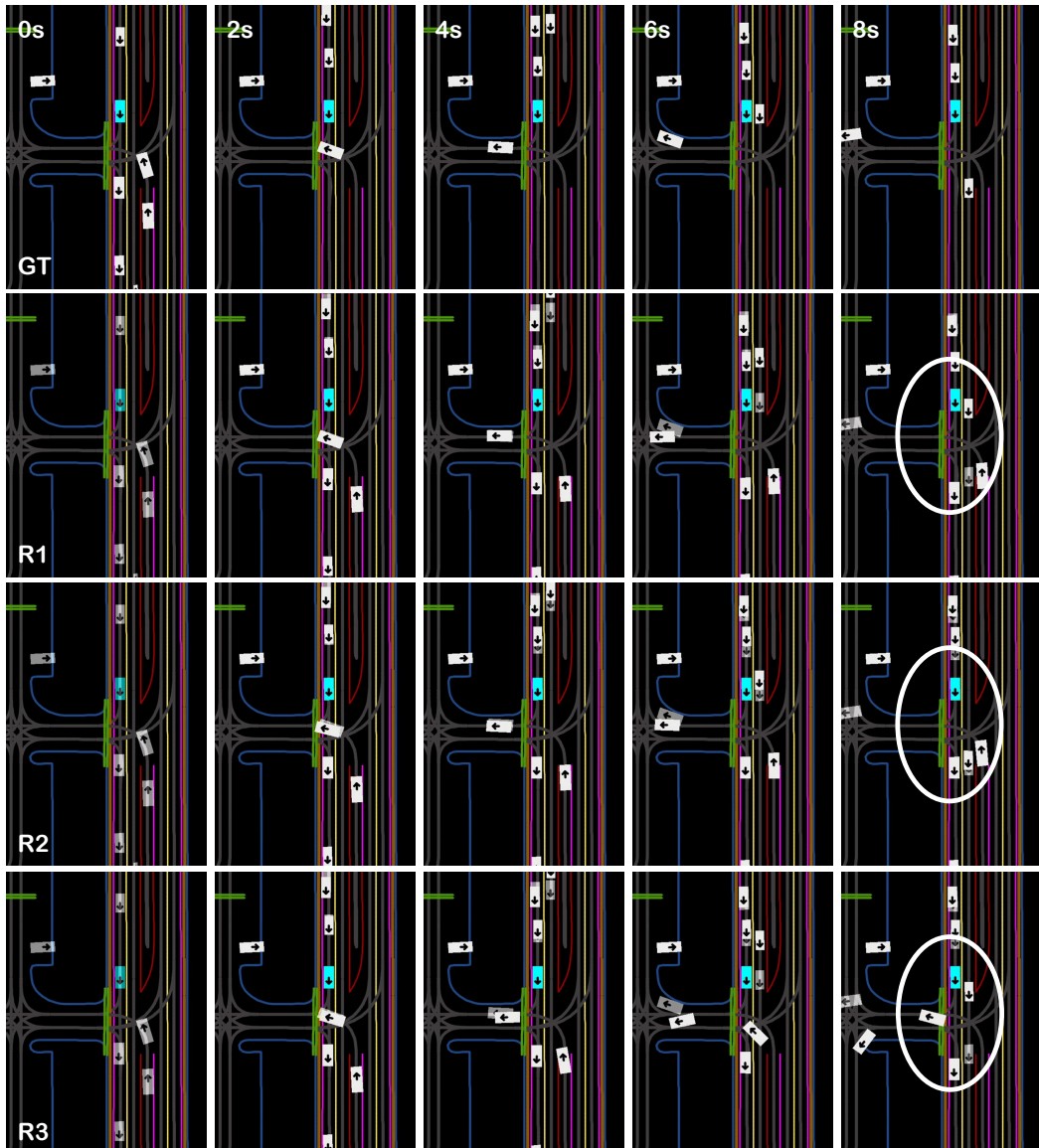

Figure 6: **Simulation results of SMART-R1 at an intersection with multi-agent negotiation, focusing on two vehicles engaged in a strong interaction.** The top row shows the ground-truth (GT) scenario, while the three rows below illustrate three representative rollouts generated by our model for the same scene. The blue box denotes the ego vehicle, and white boxes denote other traffic participants, including cars and pedestrians. Transparent boxes indicate ground-truth agent positions, whereas solid boxes represent simulated agents. The visualizations depict three distinct behaviors: (R1) conservative waiting with mutual yielding at the intersection, (R2) the left-turning vehicle yielding to the through-going vehicle, and (R3) the through-going vehicle yielding to the left-turning vehicle.

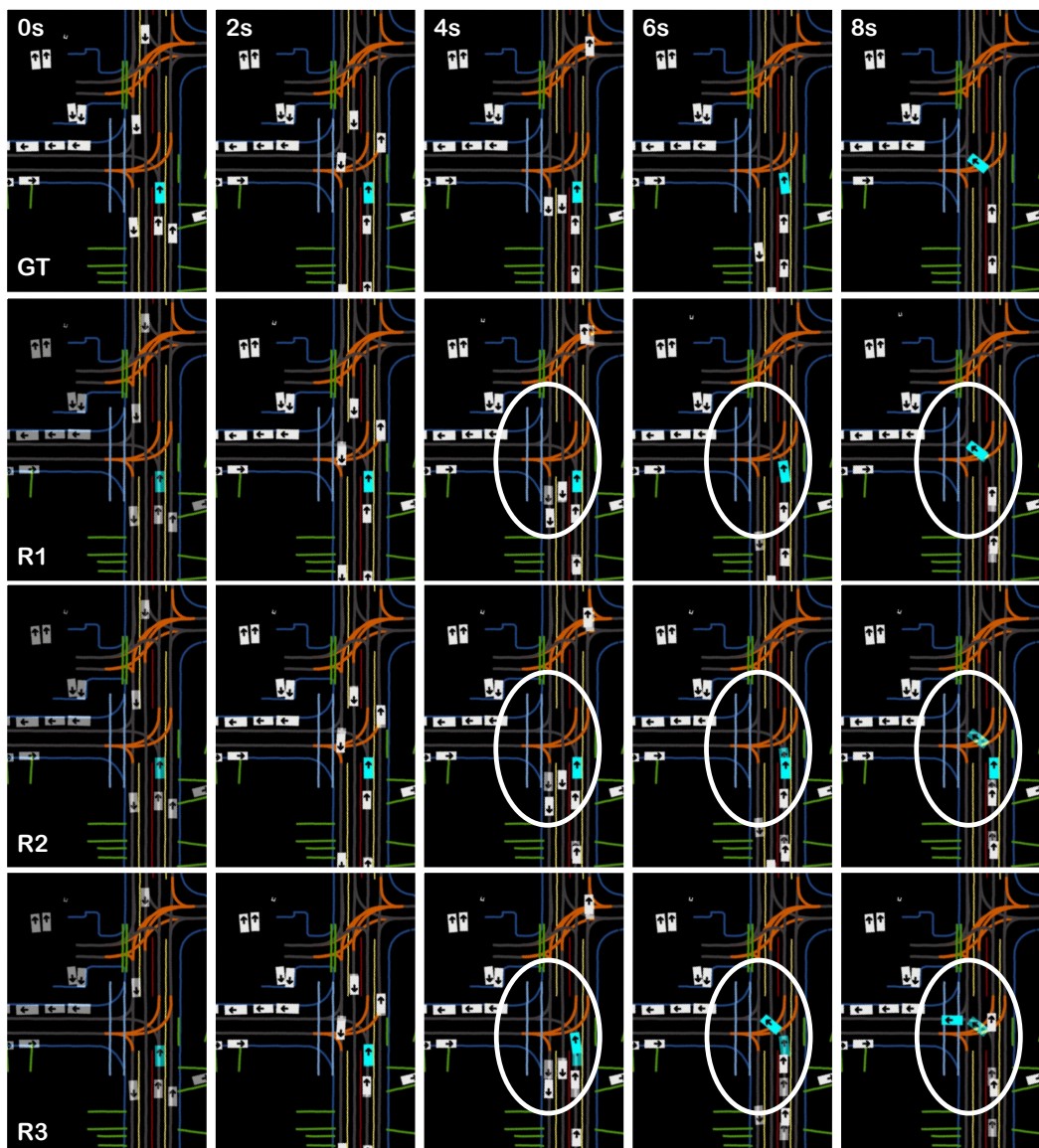

Figure 7: **Simulation results of SMART-R1 at an intersection, featuring negotiation between the ego vehicle and a traffic participant.** The top row shows the ground-truth (GT) scenario, while the three rows below illustrate three representative rollouts generated by our model for the same scene. The blue box denotes the ego vehicle, and white boxes denote other traffic participants, including cars and pedestrians. Transparent boxes indicate ground-truth agent positions, whereas solid boxes represent simulated agents. The visualizations depict three distinct behaviors: (R1) replication of the logged trajectory, (R2) a conservative maneuver, and (R3) an aggressive maneuver.

