# OpenReview forum: "Advancing Multi-agent Traffic Simulation via R1-Style Reinforcement Fine-Tuning"
_ICLR.cc/2026/Conference — ICLR 2026 Poster_

### Official Review · Reviewer_PHPM · 2025-10-17

**Soundness:** 2
**Presentation:** 3
**Contribution:** 2
**Rating:** 4
**Confidence:** 5

**Summary:**

The paper apply the R1 training style to the traffic model learning task to solve the distribution shift problem. It directly uses the waymo evaluation metrics to assign the reward. The method achives good result in the Waymo Sim Agent benchmark.

**Strengths:**

1. The performance of the method is good on the Waymo Sim Agent benchmark.
2. The paper is good written and easy to follow.
3. The ablation experiments are extensive.

**Weaknesses:**

1. The contribution of the paper is limited. It looks like an engineering work to apply the existing R1 training style to finetune an existing model based on CATK.
2. The metric-oriented policy optimization method directly uses the evaluation metric as the reward, which makes its strong performance less meaningful. It may simply overfit to the metric itself rather than improving genuine behavioral realism across other metrics.
3. Taking the waymo sim agent score as reward is time consuming because the computation is slow which requires 32 rollouts.

**Questions:**

1. How do you assign the rewad among the 32 rollout? Do all 32 rollouts use the same reward?
2. What is the performance of using the original KL penalty?
3. What is the performance of using more iterations of SFT and RFT or making them optimize together?

---

> ### Author Response · Authors · 2025-11-21
> **Response to Reviewer PHPM (1/2)**
>
> We sincerely appreciate the reviewer’s affirmative comments on the model performance, writing clarity, and extensive ablations, as well as the constructive feedback and valuable suggestions for improvement. Below, we provide detailed responses to each concern.
>
> ---
>
> > W1. The contribution of the paper is limited. It looks like an engineering work to apply the existing R1 training style to finetune an existing model based on CATK.
> >
>
> Thank you for your comments, and we would like to clarify that our contribution extends beyond engineering and includes two core conceptual advances:
>
> - First application of R1-style post-training to multi-agent traffic simulation: To the best of our knowledge, our work is the first to introduce an iterative "SFT–RFT–SFT" (R1-style) post-training pipeline directly into the domain of multi-agent traffic simulation for autonomous driving. We demonstrate that this paradigm, originally developed for LLMs, is highly effective for improving simulation realism and achieves state-of-the-art performance on the WOSAC benchmark. This represents a non-trivial insight: the R1-style structure can be generalized beyond language to interactive multi-agent traffic simulation.
> - A novel Metric-oriented Policy Optimization (MPO) strategy: Our method introduces a new policy optimization approach that leverages predictable reward expectations as prior knowledge to guide policy updates more efficiently during RFT. This differs from existing GRPO/PPO/DPO-style algorithms, and we believe it is a meaningful methodological contribution rather than a direct reuse of prior techniques.
>
> ---
>
> > W2. The metric-oriented policy optimization method directly uses the evaluation metric as the reward, which makes its strong performance less meaningful. It may simply overfit to the metric itself rather than improving genuine behavioral realism across other metrics.
> >
>
> We appreciate the reviewer’s thoughtful perspective. Our response is as follows:
>
> - The Realism Meta metric in WOSAC [1] is carefully designed by Waymo as a composite measure capturing kinematic realism, interaction plausibility, and map adherence. It provides a comprehensive and domain-validated evaluation of behavioral realism.
> - Our framework is general: if alternative, well-defined objectives are provided (e.g., human preference scores, auxiliary realism metrics), the same post-training procedure can optimize toward those objectives.
> - Importantly, optimizing toward evaluation metrics is consistent with the philosophy of learning-based systems. We agree that overfitting to the training set is undesirable, but optimizing toward the task’s training objectives is meaningful and expected.
>
> We therefore believe that optimizing toward WOSAC’s realism criteria is meaningful and aligned with the goals of traffic simulation, rather than a case of trivial metric exploitation.
>
> ---
>
> > W3. Taking the waymo sim agent score as reward is time consuming because the computation is slow which requires 32 rollouts.
> >
>
> Thank you for the insightful point regarding the computational overhead. Indeed, we agree that computing the WOSAC score across 32 rollouts is slow. To mitigate this cost while retaining training effectiveness, we only evaluate the most probable rollout and assign its score as the reward. The intuition is that encouraging the policy to increase the probability of its most realistic rollout is both computationally manageable and empirically effective to improve overall policy quality.
>
> > Q1. How do you assign the rewad among the 32 rollout? Do all 32 rollouts use the same reward?
> >
>
> Thank you for the question. As noted above, our MPO framework evaluates only the rollout with the highest likelihood and assigns its score as the reward. This significantly reduces computational overhead while guiding the model toward realistic high-probability rollouts.
>
> ---
>
> > Q2. What is the performance of using the original KL penalty?
> >
>
> We appreciate this insightful question. In policy optimization for RFT, we typically compute the per-token KL penalty using an estimator as in GRPO, for the following reasons:
>
> - The KL term cannot be computed analytically, since computing the exact KL divergence requires summing over the entire token vocabulary, which requires too much computation or memory [2].
> - In RL, we only observe a single sampled trajectory per rollout, drawn directly from the policy, rather than the full distribution. Recovering the true distribution is therefore impractical, making an estimator necessary.
> - Since the KL penalty here is used primarily to constrain deviation from the distribution of the base model, an exact value is unnecessary. Therefore, an estimator is sufficient and appropriate.
>
> As a result, we adopt the unbiased, minimum-variance estimator for the KL penalty, as described in Equation (4) of the manuscript.

---

> > ### Author Response · Authors · 2025-11-21
> > **Response to Reviewer PHPM (2/2)**
> >
> > > Q3. What is the performance of using more iterations of SFT and RFT or making them optimize together?
> > >
> >
> > We appreciate the reviewer highlighting this valuable direction. Following the multi-stage paradigm widely adopted in LLM post-training, we experimented with adding additional "SFT–RFT" cycles. These extra stages introduce substantial post-training computational overhead, yet yield small but consistent gains:
> >
> > - One additional cycle increases the Realism Meta score from 0.7858 to 0.7860;
> > - A second additional cycle further improves it to 0.7862.
> >
> > ---
> >
> > **References**
> >
> > [1] [The Waymo Open Sim Agents Challenge](https://arxiv.org/pdf/2305.12032), NeurIPS, 2023.
> >
> > [2] http://joschu.net/blog/kl-approx.html

---

> > > ### Comment · Reviewer_PHPM · 2025-11-25
> > > **Response**
> > >
> > > 1. Using metric as reward is not novel. Does MPO have any other novelty?
> > > 2. "we only evaluate the most probable rollout and assign its score as the reward". Do you use the original policy distribution for updating or the greed one? What is the performance of sampling according to the original policy. You should make it more clear in the paper. Does all agents use the same reward?
> > > 3. For the original KL penalty, I mean using the Pi_theta_ref/pi_theta only not the analytical one.
> > > 4. 0.7858 is the tested on waymo leadboard. So please publish your result of 0.7862 on this leadboard.

---

> > > > ### Comment · Reviewer_YBue · 2025-11-25
> > > >
> > > > I'd strongly recommend that the reviewer PHPM revises their comments about novelty. "applications to robotics, autonomy, planning" is one of the subject areas of ICLR. ICLR does not require each paper to have a novel method and this is not a rejection reason. I also don't think the other two negative remarks of reviewer PHPM constitute a reason to reject this work.

---

> ### Author Response · Authors · 2025-11-26
> **Response to Reviewer PHPM**
>
> Thank you very much for your prompt and thoughtful feedback. We address the follow-up questions below.
>
> ---
>
> > Using metric as reward is not novel. Does MPO have any other novelty?
> >
>
> We appreciate the opportunity to clarify this point. The novelty of our proposed MPO does not solely lie in the idea of "metric-as-reward" itself. Instead, its core contribution is to **leverage predictable reward expectations as prior knowledge** to guide policy updates **more efficiently** during RFT. This enables a simple yet effective policy optimization procedure that differs from existing approaches. For example, unlike PPO-style methods, MPO does not require training a critic network, and unlike GRPO, it does not require generating large batches of samples for policy improvement. Additionally, to the best of our knowledge, our work is also the first to explicitly employ the Realism Meta metric as the reward within the RFT setting, which is tailored specifically for the traffic simulation domain.
>
> ---
>
> > "we only evaluate the most probable rollout and assign its score as the reward". Do you use the original policy distribution for updating or the greed one? What is the performance of sampling according to the original policy. You should make it more clear in the paper. Does all agents use the same reward?
> >
>
> Thank you for raising this point. If we understand correctly, the “original policy” refers to sampling 32 rollouts and assigning the corresponding metric as the reward. In our method, we use the most probable rollout, as this most likely rollout already reflects the capability of the current model. Our empirical comparison on the validation split shows that the “original policy” (0.7738) and ours (0.7740) yield **almost identical** Realism Meta metric after RFT, while the former incurs additional computational overhead.
>
> Furthermore, all simulated agents receive the same reward because the official Realism Meta metric is defined at the **scene level**, rather than at the per-agent level.
>
> ---
>
> > For the original KL penalty, I mean using the Pi_theta_ref/pi_theta only not the analytical one.
> >
>
> Thank you for the explicit clarification. The experiment on the validation split shows that using the pi_theta_ref/pi_theta as regularization results in a Realism Meta metric of 0.7481 after RFT, which is lower than ours (0.7740). This performance drop further supports the effectiveness of the unbiased KL estimator used in our MPO design.
>
> ---
>
> > 0.7858 is the tested on waymo leadboard. So please publish your result of 0.7862 on this leadboard.
> >
>
> Thank you for the suggestion. We fully intend to publish our updated results on the WOSAC leaderboard. Our only concern is whether posting during the rebuttal phase could risk **de-anonymization**, according to the official ICLR guidelines:
>
> *"Q: Can you explain how to treat de-anonymization in the case where a submitted paper refers to a challenge they won which can identify the authors?*
>
> *It is ok to report results on a challenge leaderboard. Authors may include the ranking and the name of the challenge. ___Reviewers will be advised not to intentionally search for the authors by examining the leaderboard.___"*
>
> We are willing to publish the leaderboard entry as soon as **the ACs / PCs confirm that doing so during the rebuttal phase complies with anonymity requirements**. We will proceed immediately once approval is provided.

---

### Official Review · Reviewer_tSgk · 2025-10-27

**Soundness:** 3
**Presentation:** 3
**Contribution:** 3
**Rating:** 6
**Confidence:** 4

**Summary:**

This paper proposes to use reinforcement learning to post-train the Next-token-prediction-based motion planning model. It uses a three-stage post-training method, SFT-RLFT-SFT, and finally achieves SOTA on Waymo Open Sim Agent Challenge (WOSAC).

**Strengths:**

1. Agent simulation is an important problem for self-driving, especially building agents with realistic behaviors.
2. The paper proposes to use RL post-training to explore better behaviors to increase the realism and achieve SOTA on the commonly used benchmark WOSAC. Comprehensive experiment results, including ablations, demonstrate the effectiveness of the proposed method.
3. The paper writing is good and easy to understand.

**Weaknesses:**

1. The method is mainly designed for an NTP-based model, while new adaptations are needed for continuous prediction models like diffusion models, which may limit its future application.
2. It would be better to evaluate the method on more end-to-end planning benchmarks like nuPlan to show its general capabilities, which can further increase its impact.

**Questions:**

N/A

---

> ### Author Response · Authors · 2025-11-21
> **Response to Reviewer tSgk**
>
> We sincerely appreciate the reviewer’s thoughtful and encouraging feedback. We are grateful for the recognition of the importance of the problem, the strength of our experimental results, and the clarity of the writing. We also thank you for the constructive suggestions, which help illuminate valuable directions for future work. Below, we address each comment in detail.
>
> ---
>
> > **W1.** The method is mainly designed for an NTP-based model, while new adaptations are needed for continuous prediction models like diffusion models, which may limit its future application.
> >
>
> Thank you for the insightful feedback. Current multi-agent traffic simulators largely follow two dominant paradigms: Next-Token-Prediction (NTP) models and diffusion-based models. Diffusion-based approaches often incur significant computational overhead and struggle with complex multi-agent interactions, which have limited their competitiveness on large-scale simulation benchmarks. Consequently, recent advances in the field have increasingly converged toward the NTP pipeline. Our approach builds upon this prevailing trend. Since NTP models share commonalities with autoregressive LLMs, they naturally support post-training strategies. Our goal is to show that such an R1-style post-training procedure, proven effective in LLM reasoning (e.g., DeepSeek-R1), can be fruitfully adapted to large-scale interactive traffic simulations for behavior preference alignment and improving simulation realism. We agree that extending post-training paradigms to diffusion-based simulators is a promising avenue for future research; however, such exploration would require substantial methodological adjustments and fall outside the scope of the present work.
>
> ---
>
> > **W2.** It would be better to evaluate the method on more end-to-end planning benchmarks like nuPlan to show its general capabilities, which can further increase its impact.
> >
>
> We appreciate the suggestion to evaluate our method on planning benchmarks. However, we respectfully note that multi-agent traffic simulation and motion planning differ fundamentally in both problem formulations and evaluation objectives:
>
> - Multi-agent traffic simulation aims to model the joint distribution of future trajectories for all agents in the scene, including both the ego vehicle and surrounding traffic participants, so as to simulate realistic, scene-level multi-agent traffic behaviors.
> - Motion planning, in contrast, focuses exclusively on generating a safe and feasible future trajectory for the ego vehicle, typically conditioned on route-level navigation inputs or driving-goal signals that are not available in the simulation tasks.
>
> Our framework treats all agents uniformly and is designed to model joint behaviors rather than plan for a designated ego agent. Consequently, it does not incorporate planning-specific elements such as navigation goals, route preferences, or ego-centric cost functions. For these reasons, the current simulator design is not directly aligned with existing end-to-end planning benchmarks such as nuPlan. While we believe that adapting our architecture (e.g., by incorporating routing information and modifying the training objective to support ego-centric planning) could enable evaluation on planning benchmarks, such extensions are beyond the scope of this work. We fully agree, however, that this is a valuable and promising direction for future research.

---

> > ### Comment · Reviewer_tSgk · 2025-11-25
> > **Thank you for the response**
> >
> > Thank you for your response. I would like to keep my score unchanged.

---

> > > ### Author Response · Authors · 2025-11-26
> > >
> > > We would like to express our sincere gratitude for your time and constructive feedback in strengthening our work. We truly appreciate your positive assessment and encouraging support.

---

### Official Review · Reviewer_uZAq · 2025-11-01

**Soundness:** 3
**Presentation:** 3
**Contribution:** 2
**Rating:** 2
**Confidence:** 4

**Summary:**

- This paper tries to address the problem of distribution shift in the learned (supervised learning) simulators affecting their generalization capabilities.
- The authors propose SMART-R1 to align agent behavior with human preferences and relevant evaluation metrics for agent traffic simulation tasks.
- The authors introduce an iterative (interleaving) SFT-RFT-SFT post-training pipeline to maximize the performance and minimize the skills learned in previous training runs (catastrophic forgetting).
  - They introduce Metric-oriented Policy Optimization (MPO) to exploit the knowledge from priors and can try to optimize for collision, off-route cases, etc. that might not be there in the SFT data.
- With the above recipe the proposed method was able achieve best performance on Waymo Open Sim Agents Challenge (WOSAC)

Given the comments related to weaknesses and limitations, I can have the rating as 2. Flexible to move this during or after rebuttal.

**Strengths:**

- The authors presented the text, figures and tables nicely.
- Converting the trajectories into motion tokens makes it easier for the sequence to be fed to the transformer and formulate it as a next-token-prediction (NTP) task.
- The closed-loop SFT seems to reduce the covariate shift in NTP models leading to improved performance.
- RFT stage aligns the model with metric preferences and enhances realism.
- The paper does a good job in performing experiments and ablations to study the effect KL regularization, effect of different post-fine-tuning methods, effects of using different RL algorithms for RFT.

**Weaknesses:**

The approach proposed has off-late been a pretty common fine-tuning strategy for many robotics tasks. It is appreciated that the authors have done research in incorporating the multi-step post-training pipeline serving a dual purpose of maximizing performance and minimizing forgetting. However, there has been some work done on dealing with forgetting as well in LLMs ([Towards a Unified View of Large Language Model Post-Training](https://arxiv.org/pdf/2509.04419)) .

**Questions:**

1. Was keeping a KL term in the SFT tried?

2. Was there any KL term in the RL loss to minimize the divergence from the base policy while maximizing returns?

3. Was any other regularizer tried to prevent forgetting (for example, Elastic Weight Consolidation)?

4. Was there any KL term in the second SFT training (refer to [RL's Razor: Why Online Reinforcement Learning Forgets Less](https://arxiv.org/abs/2509.04259))?

5. How many agents are considered in the scenario and how are selected or filtered, if any?

6. Line 397: “Notably, applying two consecutive SFT phases” - isn't this just one big SFT stage with 2x epochs?

---

> ### Author Response · Authors · 2025-11-21
> **Response to Reviewer uZAq**
>
> We sincerely appreciate the reviewer’s thoughtful assessment of our work, including the recognition of our research contributions, writing clarity, and comprehensive experimental study. We also thank the reviewer for the constructive feedback and valuable suggestions. Below, we address each concern in detail.
>
> ---
>
> > **W1.** The approach proposed has off-late been a pretty common fine-tuning strategy for many robotics tasks. It is appreciated that the authors have done research in incorporating the multi-step post-training pipeline serving a dual purpose of maximizing performance and minimizing forgetting. However, there has been some work done on dealing with forgetting as well in LLMs ([Towards a Unified View of Large Language Model Post-Training](https://arxiv.org/pdf/2509.04419)).
> >
>
> Thank you for acknowledging our work on applying a multi-stage R1-style post-training pipeline to traffic simulation in autonomous driving. We also appreciate your pointer to the excellent concurrent submissions to the ICLR 2026 venue [1, 2], which provide strong theoretical insights into forgetting in LLM post-training. To the best of our knowledge, our work is the first to explore such an iterative "SFT–RFT–SFT" strategy in the domain of multi-agent traffic simulation for autonomous driving. Empirically, we observe that adding an additional SFT stage after RFT helps the model better balance metric-driven optimization with the preservation of the behavioral distribution learned during the initial SFT. This leads to notable improvements in simulation realism. We believe the effectiveness of this procedure is closely tied to its ability to mitigate forgetting, which is an observation that aligns surprisingly well with emerging theoretical analyses in the LLM literature. Although our study is focused on applications to autonomous driving, our empirical findings offer complementary practical evidence supporting these theoretical trends, suggesting that mitigation of forgetting is both theoretically meaningful and broadly applicable.
>
> ---
>
> > **Q1.** Was keeping a KL term in the SFT tried?
> >
>
> Thank you for the question. Both SFT stages optimize the standard cross-entropy loss, which is equivalent to minimizing the KL divergence between the empirical token distribution and the model predictions. No additional KL term is introduced beyond this.
>
> > **Q2.** Was there any KL term in the RL loss to minimize the divergence from the base policy while maximizing returns?
> >
>
> Yes. As shown in Equation (5) of the manuscript, the RFT objective explicitly includes a KL term to restrict deviation from the reference policy.
>
> > **Q3.** Was any other regularizer tried to prevent forgetting (for example, Elastic Weight Consolidation)?
> >
>
> Thank you for the question. Our work focuses on the post-training pipeline in the context of autonomous driving and follows commonly adopted and effective practices from LLM post-training. We did not incorporate additional mechanisms. Instead, we rely on an additional SFT stage to "re-anchor" the policy to the logged behavior distribution after RFT, which in practice helps mitigate forgetting from policy updates.
>
> > **Q4.** Was there any KL term in the second SFT training (refer to [RL's Razor: Why Online Reinforcement Learning Forgets Less](https://arxiv.org/abs/2509.04259))?
> >
>
> Thank you for the question. The second SFT stage again optimizes cross-entropy, which indirectly minimizes KL, and serves to stabilize and refine the policy after RFT updates. No additional KL term is introduced in the second SFT training.
>
> ---
>
> > **Q5.** How many agents are considered in the scenario and how are selected or filtered, if any?
> >
>
> Thank you for the question. We follow the official setup of the Waymo Open Sim Agents Challenge (WOSAC) [3]. Each scenario contains up to 128 agents. An agent is included for simulation only if it has valid logged measurements at the initial time step (i.e., the last logged step before simulation begins).
>
> ---
>
> > **Q6.** Line 397: “Notably, applying two consecutive SFT phases” - isn't this just one big SFT stage with 2x epochs?
> >
>
> Thank you for raising this point. Your interpretation is correct: applying two consecutive SFT phases corresponds to training for twice the number of SFT epochs. We phrase it this way in the ablation context (Table 3) to clearly illustrate that without an RFT phase interleaved between the two SFT stages, performance degrades. This expression helps highlight the role of RFT in the multi-stage pipeline.
>
> ---
>
> **References**
>
> [1] [Towards a Unified View of Large Language Model Post-Training](https://openreview.net/forum?id=3UxhlRCVgr), ICLR 2026 submission.
>
> [2] [RL's Razor: Why Online Reinforcement Learning Forgets Less](https://openreview.net/forum?id=7HNRYT4V44), ICLR 2026 submission.
>
> [3] [The Waymo Open Sim Agents Challenge](https://arxiv.org/pdf/2305.12032), NeurIPS, 2023.

---

### Official Review · Reviewer_YBue · 2025-11-06

**Soundness:** 4
**Presentation:** 3
**Contribution:** 3
**Rating:** 8
**Confidence:** 3

**Summary:**

The paper introduces a trajectory forecasting system that is based on SMART and adds DeepSeek-R1-style training, which ultimately leads to SotA performance on a difficult dataset.

**Strengths:**

- **S.1:** The idea is simple and well-justified. It's not methodologically revolutionary but the results are very impressive.
- **S.2:** The writing is clear, the illustrations are simple but good.
- **S.3:** I really like your ablations, especially table 3-6. I think this is very good scientific practice.

**Weaknesses:**

- **W.1:** Reproducibility. I really dislike when papers don't submit any code with their submission for a system that's quite complex. Especially since your method is built on top of an existing codebase, I think it would've been great if you could've included an anonymous github link or a zip file with the submission.
- **W.2:** Performance/Train time? Along the same lines of the previous comment: how long was this trained, on which hardware? How fast is inference?
- **W.3:** The writing is moooostly clear but there are some sharp edges that I think need to be improved. In the abstract, you already mentioned "R1-style..." without mentioning what this is referring to (e.g."the SFT-RL-SFT training regimen of DeepSeek-R1") and you often refer to CAT-K rollouts, which I'm not familiar with and since it seems important to your method, it'd be nice to have a quick explanation in the main body of your paper.

**Questions:**

n/a

---

> ### Author Response · Authors · 2025-11-21
> **Response to Reviewer YBue**
>
> We sincerely thank the reviewer for the thoughtful and encouraging comments. We greatly appreciate the recognition of the paper’s soundness, clarity, and empirical contributions, as well as the constructive suggestions for improvement. Below, we provide detailed responses to each point.
>
> ---
>
> > **W1.** Reproducibility. I really dislike when papers don't submit any code with their submission for a system that's quite complex. Especially since your method is built on top of an existing codebase, I think it would've been great if you could've included an anonymous github link or a zip file with the submission.
> >
>
> We would very much like to open-source our code; however, because this project was conducted in collaboration with an industry partner, the release must undergo a mandatory internal review and approval process that is beyond our control. We will make the repository publicly accessible once we receive the necessary approvals.
>
> ---
>
> > **W2.** Performance/Train time? Along the same lines of the previous comment: how long was this trained, on which hardware? How fast is inference?
> >
>
> Thank you for requesting additional information regarding resource usage. The complete training pipeline takes approximately 39 hours on 64 NVIDIA H20 GPUs. At inference time, the model achieves an average latency of about 15 ms per simulation step using a standard PyTorch implementation.
>
> ---
>
> > **W3.** The writing is moooostly clear but there are some sharp edges that I think need to be improved. In the abstract, you already mentioned "R1-style..." without mentioning what this is referring to (e.g."the SFT-RL-SFT training regimen of DeepSeek-R1") and you often refer to CAT-K rollouts, which I'm not familiar with and since it seems important to your method, it'd be nice to have a quick explanation in the main body of your paper.
> >
>
> We appreciate the reviewer’s careful reading and constructive feedback on clarity. We have revised the manuscript to provide clearer descriptions accordingly.

---

> > ### Comment · Reviewer_YBue · 2025-11-25
> >
> > Thanks for the responses. I'm staying with my current rating - I think this should be accepted, but I don't think it's one of the best papers at ICLR. I've updated my confidence.
> >
> > I also strongly suggest the other reviewers dial down their negative remarks of lack of novelty. Applying a method to an entirely new problem is well within spec of ICLR.

---

> > > ### Comment · Reviewer_PHPM · 2025-11-26
> > > **Response**
> > >
> > > 1. Applying RL funetuning to traffic simulation is not new [1]. Applying a method to an entirely new problem is ok. But there exitsts similar previous work (missed in the paper) in the problem and applying the method leads very limited improvement compared to previous method.
> > > 2. CAT-K is a well-known finetuning method for traffic simulation but you are unfamiliar with. If you are not an expert in traffic simulation, please lower your confidance.
> > >
> > >
> > > [1] Peng Z, Luo W, Lu Y, et al. Improving agent behaviors with rl fine-tuning for autonomous driving[C]//European Conference on Computer Vision. Cham: Springer Nature Switzerland, 2024: 165-181.

---

> ### Author Response · Authors · 2025-11-26
> **Response to Reviewer YBue**
>
> We would like to express our sincere gratitude for your time, constructive feedback, and encouraging support. We truly appreciate your positive assessment and your recognition of the contribution our work makes.

---

> ### Author Response · Authors · 2025-11-26
> **Response to Reviewer PHPM**
>
> We sincerely thank Reviewer **PHPM** for the follow-up comments and the opportunity to clarify our contribution.
>
> ---
>
> > Applying RL funetuning to traffic simulation is not new [1]. Applying a method to an entirely new problem is ok. But there exitsts similar previous work (missed in the paper) in the problem and applying the method leads very limited improvement compared to previous method.
> >
>
> We respectfully highlight two key distinctions from Ref. [1]:
>
> 1. Our work introduces a full **R1-style post-training** framework that alternates iterative “**SFT–RFT–SFT”** stages. This multi-stage pipeline is fundamentally different from Ref. [1], which applies only a single RFT process. The full post-training procedure proves highly effective for enhancing simulation realism, enabling our model to achieve **state-of-the-art performance on the WOSAC benchmark**.
> 2. We propose a simple yet effective **Metric-oriented Policy Optimization (MPO)** strategy that differs from existing approaches (e.g., PPO, GRPO). In contrast, Ref. [1] employs the standard REINFORCE algorithm with a simple reward design. Our MPO method is tailored for preference alignment in multi-agent traffic simulation and constitutes a core contribution of this work.
>
> ---
>
> > CAT-K is a well-known finetuning method for traffic simulation.
> >
>
> We fully acknowledge that CAT-K is a strong and well-recognized method, and we do not claim it as part of our contributions. Our contributions instead lie in: (i) the **first application of an R1-style post-training pipeline to multi-agent traffic simulation**, (ii) the development of a **new MPO strategy** tailored to this domain, and (iii) the achievement of **state-of-the-art performance**, which we believe highlights that R1-style post-training, originally developed for LLM reasoning (e.g., DeepSeek-R1), can be effectively adapted to large-scale autonomous driving to enhance the realism of multi-agent traffic simulations.
>
> ---
>
> **Reference**
>
> [1] Improving Agent Behaviors with RL Fine-tuning for Autonomous Driving, ECCV, 2024.

---

### Author Response · Authors · 2025-12-03
**Summary of the Rebuttal**

We sincerely thank all reviewers for their time, thoughtful evaluation, and constructive feedback. Below, we provide a brief and focused summary of the strengths consistently recognized across reviews, our responses to the raised concerns, and the outcomes of the discussion phase.

- **Broadly Recognized Strengths.**

    We appreciate that all reviewers acknowledged that our proposed R1-style post-training paradigm achieves **state-of-the-art performance on the challenging WOSAC benchmark**, supported by extensive ablations and a comprehensive experimental study. All reviewers also commended the **clarity, organization, and overall quality of the manuscript** (**Reviewers** **YBue, uZAq, tSgk, PHPM**). Several further emphasized that the method is **simple, effective, and well-motivated** (**Reviewers** **YBue, uZAq, tSgk**).

- **Point-by-Point Responses to Concerns.**

    During the discussion phase, we provided detailed, point-by-point clarifications addressing all raised questions. We summarize the key aspects of the reviewers’ feedback below:

    - For **Reviewer YBue**, we clarified issues related to reproducibility, training details, and writing clarity.
    - For **Reviewer tSgk**, we discussed the broader applicability of our paradigm beyond next-token prediction and elaborated on differences between simulation and planning tasks.
    - For **Reviewer uZAq**, we further explained our distinctions from concurrent [ICLR 2026 submissions](https://openreview.net/forum?id=3UxhlRCVgr) on theoretical forgetting in LLMs, and clarified the rationale and implementation of the KL regularization term in post-training.
    - For **Reviewer PHPM**, we elaborated on our contributions and novelty, reward assignment, and provided additional experiments on the KL estimator as well as further “RFT–SFT” iterations.
- **Discussion Outcomes.**

    We are encouraged that **Reviewers** **YBue and** **tSgk** acknowledged our responses without further concerns. In addition, we responded comprehensively to all questions from **Reviewer** **PHPM**, during which **Reviewer** **YBue** also engaged in the discussion and expressed supportive perspectives on our work. Unfortunately, we did not receive any engagement from **Reviewer** **uZAq** before the unexpected incident.


Once again, we extend our sincere gratitude to all reviewers for their time, effort, and constructive feedback.

---

### Meta-Review · Area_Chair_g85H · 2026-01-04

**Summary:**

Across the four reviews, there is clear agreement that the submission achieves very strong empirical performance on the Waymo Open Sim Agents Challenge (WOSAC), with multiple reviewers describing the approach as simple, well-motivated, and supported by an unusually thorough ablation suite. The paper proposes an R1-style, multi-stage post-training recipe for an NTP-based traffic-simulation model, interleaving SFT and RL fine-tuning, and introduces a metric-oriented policy optimization (MPO) strategy that directly targets the benchmark’s realism-oriented objective. The primary concerns raised in the reviews center on (i) the degree of novelty relative to existing RL fine-tuning and established simulation pipelines (e.g., CAT-K and prior “RL fine-tuning for autonomous driving” work), (ii) the risk that “metric-as-reward” could overfit to the benchmark score rather than improving broader behavioral realism, (iii) reproducibility and implementation transparency (code availability; training/inference cost), and (iv) scope/generalization beyond NTP-based simulators and beyond WOSAC (e.g., diffusion-style simulators or planning benchmarks such as nuPlan). The rebuttal addresses several decision-relevant items with concrete clarifications (compute, rollout/reward assignment, KL regularization, and additional SFT–RFT iterations) and resolves multiple presentation questions, though limitations about broader generalization and code release remain largely scope- or logistics-driven rather than newly evidenced.

**Reviewer Concerns:**

Concerns addressed by the rebuttal:
1. Reproducibility and resource reporting: The authors provided explicit training and inference cost details (training wall-clock time and hardware scale; inference latency per simulation step) and explained constraints on code release due to an industry-partner approval process, committing to release once permitted.
2. Method clarity and pipeline description: The rebuttal clarified what is meant by “R1-style” training in this context (iterative SFT-RFT-SFT), and responded to requests for clearer definitions (e.g., CAT-K rollouts) and training-stage distinctions. The authors also clarified that two consecutive SFT phases are equivalent to more SFT epochs, and highlighted via ablations that interleaving RFT is the key ingredient.
3. Reward assignment and rollout handling: In response to concerns that the benchmark reward is expensive to compute (32 rollouts), the authors clarified their training procedure: they score only the most probable rollout and use its score as the reward, reporting a validation comparison indicating similar post-RFT performance to scoring all rollouts but at substantially lower compute. They also clarified that the WOSAC metric is scene-level, so all agents share the same reward signal.
4. KL regularization and estimator choice: The rebuttal clarified the use of a KL term in the RFT objective and provided an additional comparison indicating that a naïve alternative KL regularization performs materially worse than the chosen (unbiased/minimum-variance) estimator, supporting the MPO design choice. Additional experiments on the KL estimator were also described.
5. Additional training iterations: The authors reported that adding more SFT–RFT cycles yields small but consistent gains, strengthening the empirical characterization of the proposed post-training recipe.

Concerns partially addressed or still outstanding:
1. Novelty relative to prior RL fine-tuning and existing simulation pipelines: One reviewer argues the paper is primarily an application/engineering effort and cites prior work on RL fine-tuning for autonomous driving. The rebuttal positions novelty in the full multi-stage post-training recipe and in MPO’s update design. However, the perceived novelty remains reviewer-dependent and would benefit from clearer positioning and explicit comparisons/discussion of the closest prior work in the final version.
2. “Metric-as-reward” and generalization beyond the benchmark metric: The rebuttal argues that the WOSAC Realism Meta metric is a carefully designed composite and that optimizing it is aligned with the task objective; however, concerns about cross-metric generalization (or whether improvements transfer to alternative realism criteria) are not fully resolved with new evidence in the material provided.
3. Scope beyond NTP-based simulation and beyond WOSAC: A reviewer notes that the method is tightly coupled to NTP-based models and asks for broader evaluation (e.g., diffusion-based simulators or end-to-end planning benchmarks). The rebuttal explains why such extensions are non-trivial and frames them as future work; no new experiments in these settings are presented.

**Reviewer Scores:**

Reviewer YBue: Likely unchanged and supportive. The reviewer explicitly maintains an accept recommendation after the authors' responses, emphasizing strong results and solid ablations, while encouraging other reviewers to moderate novelty-related objections.

Reviewer tSgk: Likely unchanged. The reviewer’s concerns are framed as scope/generalization requests (diffusion models; planning benchmarks). The rebuttal clarifies the scope distinction between simulation and planning and notes diffusion extensions as future work; the reviewer explicitly keeps the score.

Reviewer PHPM: Unclear; potentially modestly higher on clarity, but novelty concerns persist. The rebuttal provides specific answers on rollout reward assignment, KL regularization design, and additional SFT–RFT cycles, and clarifies MPO’s intended novelty (critic-free and sample-efficient updates). However, the reviewer continues to emphasize novelty and metric-overfitting concerns and requests stronger public leaderboard evidence, so a substantial upward score change is uncertain.

Reviewer uZAq: Likely increase slightly. The rebuttal addresses the reviewer’s questions directly (KL usage, agent selection, interpretation of multiple SFT phases) and clarifies the relation to concurrent forgetting work.

---

### Decision · Program_Chairs · 2026-01-26

Accept (Poster)